# Assessing the Impact of Spraying an *E. faecium* Probiotic at Hatch and Supplementing Feed with a Triple-Strain *Bacillus*-Based Additive on BCO Lameness Incidence in Broiler Chickens

**DOI:** 10.3390/ani15121765

**Published:** 2025-06-15

**Authors:** Khawla Alharbi, Anh Dang Trieu Do, Abdulaziz Alqahtani, Ruvindu Perera, Alexa Thomas, Antoine Meuter, Adnan Ali Khalaf Alrubaye

**Affiliations:** 1Cell and Molecular Biology Program, University of Arkansas, Fayetteville, AR 72701, USA; ka030@uark.edu (K.A.); ad086@uark.edu (A.D.T.D.); aqahtani@uark.edu (A.A.); rperera@uark.edu (R.P.); 2Center of Excellence for Poultry Science, University of Arkansas, Fayetteville, AR 72701, USA; alt032@uark.edu; 3Dale Bumpers College of Agricultural, Food, and Life Sciences, University of Arkansas, Fayetteville, AR 72701, USA; 4Animal and Plant Health & Nutrition, Novonesis, 2970 Hørsholm, Denmark; antme@novonesis.com

**Keywords:** lameness, broilers, BCO, probiotic, bone

## Abstract

Bacterial chondronecrosis with osteomyelitis (BCO) is a major contributor to lameness in broiler chickens, often resulting from bacteria such as *Enterococcus*, *Staphylococcus*, and *E. coli* entering the bloodstream and colonizing bone tissue. Rapid growth, high body weight, and poor flooring and nutrition cause increased BCO risk, leading to pain, immobility, and welfare issues. This study evaluated the effects of an *Enterococcus faecium*-based spray at hatch and a *Bacillus*-based feed supplement, applied individually or together, on reducing BCO lameness. A wire-flooring challenge model was used to mimic bacterial exposure in commercial settings. All probiotic treatments significantly lowered lameness compared to untreated controls. The combined treatment was most effective, reducing lameness by nearly 48%. These findings support a promising, antibiotic-free approach to improving broiler health and welfare.

## 1. Introduction

Skeletal disorders in broilers remain a significant challenge for the poultry industry, with considerable implications for both animal welfare and economic sustainability [1]. Modern broiler strains have been intensively selected for rapid growth and high meat yield, but these improvements in productivity have come at the cost of skeletal integrity, particularly in long bones such as the femur and tibia [2]. Rapid body weight gain can outpace normal bone development, increasing the risk of skeletal abnormalities such as tibial dyschondroplasia (TD), rickets, and lameness. These conditions can significantly compromise bird mobility and overall health [2,3,4]. Among the most severe forms of skeletal-related lameness is bacterial chondronecrosis with osteomyelitis (BCO), a condition marked by cartilage necrosis and subsequent bone infection. BCO significantly impairs locomotion and causes poor welfare outcomes [2,4]. The condition is often triggered by mechanical stress on the skeletal system during periods of rapid growth, which can lead to the formation of osteochondrotic clefts in the epiphyseal and physeal regions of the bone, particularly in the femoral and tibial heads [2,4]. These structural weaknesses, combined with environmental stressors and suboptimal nutrition, increase the likelihood of bacterial translocation into compromised bone tissue.

Understanding the multifactorial nature of BCO requires examining the interactions between genetics, nutrition, and management. While genetic selection has primarily focused on carcass traits, it has frequently overlooked skeletal robustness, highlighting the need for more balanced breeding programs that prioritize both performance and structural health [2,5]. Moreover, improved nutritional strategies, growth management, and environmental optimization are essential components in reducing the incidence and severity of BCO and related lameness conditions [6,7]. Preventative strategies, such as routine health monitoring and early intervention, remain vital for minimizing skeletal disorders and ensuring long-term sustainability in broiler production systems [1,5].

Given the ongoing challenges posed by skeletal disorders such as bacterial chondronecrosis with osteomyelitis (BCO), there is growing interest in nutritional and microbial strategies that support broiler health from within. Among these, probiotics have gained considerable attention as viable alternatives to antibiotic growth promoters, particularly as the industry shifts away from routine antibiotic use due to concerns over antimicrobial resistance. Probiotics, characterized as viable microbial supplements, have been extensively investigated for their capacity to modulate intestinal microbiota composition, facilitate nutrient assimilation, and enhance immunological function in poultry when administered in efficacious doses [8,9]. These functions are especially important in modern broiler production, where birds experience rapid growth and are highly susceptible to both intestinal and systemic stressors [10,11,12].

Increasing evidence suggests that gut health plays an important role not only in digestive efficiency but also in skeletal development [7]. This is particularly relevant for fast-growing broilers, where imbalances in nutrient metabolism and inflammation may predispose birds to leg disorders such as BCO. By promoting digestive health and reducing systemic stress, probiotics like *E. faecium* and Bacillus strains may help protect bone integrity and reduce the incidence of BCO [3,13,14]. At the molecular level, probiotics support bone health through a range of interconnected mechanisms. They can enhance the expression of genes involved in bone formation and mineralization, while helping to suppress inflammatory cytokines that contribute to bone degradation [15]. For example, reductions in circulating levels of pro-inflammatory markers such as interleukins and tumor necrosis factor-alpha (TNF-α) have been observed in response to probiotic supplementation, both of which are known to play a role in bone and joint disease [16,17]. Probiotics also help shape the gut microbiota in ways that increase the production of short-chain fatty acids, compounds with known anti-inflammatory properties. Together, these effects may contribute to a more balanced immune environment that helps protect bone integrity in birds challenged with conditions like BCO [18]. Their inclusion in broiler diets thus offers a dual benefit: supporting performance while also addressing key welfare and economic concerns related to skeletal health.

Two probiotic types that have shown particular promise in poultry are *Enterococcus faecium* and Bacillus-based formulations [12,13,14,15,16,17,18,19]. *E. faecium* is known for its capacity to stabilize the gut environment, inhibit pathogenic bacteria, and improve intestinal morphology, contributing to better feed conversion and growth [8,19]. Bacillus probiotics, particularly when delivered as multi-strain blends, provide complementary benefits. These include the production of digestive enzymes, improved nutrient bioavailability, and support for gut barrier function [12,20]. Bacillus strains also exert anti-inflammatory effects, contributing to overall health and improved performance [21,22]. Their use has been linked to gains in weight gain efficiency and greater resilience to disease [12,20].

Based on the known benefits of *Enterococcus faecium* and Bacillus-based probiotics in enhancing gut health and immune function, we hypothesized that targeted probiotic supplementation could reduce the incidence and severity of BCO-related lameness in broilers. Therefore, the aim of this study was to evaluate the efficacy of a probiotic program consisting of an *E. faecium*-based spray administered at hatch (GALLIPRO^®^ Hatch, Novozymes, Hørsholm, Denmark) and a triple-strain Bacillus-based feed additive (*B. subtilis* 597, *B. subtilis* 600, and *B. amyloliquefaciens* 516) (GalliPro^®^ Fit, Novozymes, Hørsholm, Denmark) provided from day 1 to day 56. These treatments were applied individually and in combination using a wire-flooring challenge model designed to simulate the transmission and progression of BCO under commercial-like conditions. This study assessed the impact of these interventions on lameness incidence, lesion development, and performance outcomes, with the goal of identifying sustainable strategies to improve skeletal health and welfare in broiler chickens.

## 2. Materials and Methods

### 2.1. Probiotic Preparation

#### 2.1.1. GalliPro^®^ Fit

A commercially available, triple-strain, Bacillus-based probiotic, GalliPro^®^ Fit (Novozymes, Hørsholm, Denmark), was used in this study. The formulation contains two Bacillus subtilis strains (*B. subtilis* 597 and *B. subtilis* 600) and one *Bacillus amyloliquefaciens* strain (*B. amyloliquefaciens* 516), and is provided as a powdered product with a guaranteed viable count of 1.6 × 10^10^ CFU/g. It was incorporated directly into the feed at a rate of 500 g per metric ton of feed, following the manufacturer’s recommended dosage. Feed was prepared every two weeks to ensure probiotic viability and uniform distribution. Birds received the probiotic continuously in their feed from day 1 to day 56 of age.

#### 2.1.2. GalliPro^®^ Hatch

The *Enterococcus faecium* strain (*E. faecium* 669) utilized in this study was derived from the commercial probiotic GalliPro^®^ Hatch (Novozymes, Hørsholm, Denmark), which contains a concentration of 2.0 × 10^11^ CFU/g. Following the manufacturer’s guidelines, a dose delivering 2.0 × 10^9^ CFU per chick was prepared and administered on day 0 via a custom-built static spray system. Chicks were arranged in groups of 60 and manually sprayed in several passes to ensure uniform coverage, with each group receiving 75 mL of the probiotic suspension. A non-toxic blue food dye was incorporated into the mixture to facilitate visual confirmation of even distribution during application.

### 2.2. Animals and Facility

This study was conducted in accordance with the guidelines approved by the University of Arkansas Institutional Animal Care and Use Committee (Protocol #23067). The experiment included 1560-dayold Cobb 500 broiler chicks (Cobb-Vantress, Inc., Siloam Springs, AR, USA), which were randomly allocated to 26 floor pens (1.5 m × 3.0 m each), resulting in an initial stocking density of approximately 750 cm^2^ per chick. On day 14, bird numbers were adjusted to 50 chicks per pen to maintain a final density of 900 cm^2^ per chick. Pens were arranged in two rows of 13 pens and randomly assigned to treatment groups. The environmentally controlled facility was equipped with automated systems to regulate temperature, lighting, and ventilation. Tunnel ventilation and evaporative cooling pads were used to maintain optimal environmental conditions. A 23L:1D photoperiod was implemented, with light intensity maintained at 20 lux throughout the trial. Environmental temperatures were adjusted incrementally throughout the trial: 32.2 °C (days 1–3), 31 °C (days 4–6), 29 °C (days 7–10), 26 °C (days 11–14), and maintained at 23 °C from day 15 onward. Each pen was equipped with a dedicated water line (connected to municipal supply) and two feeders. Prior to bird placement, water lines were flushed with diluted bleach solution (Clorox, The Clorox Company, Oakland, CA, USA) to ensure hygienic starting conditions.

### 2.3. BCO Lameness Challenge Setup

To evaluate the impact of dietary treatments on the incidence of BCO lameness, an aerosol transmission challenge model was employed, adapted from previously established protocols [23]. Two wire-floor pens were designated as infection source areas (“seeder bird” pens) and strategically placed near the evaporative cooling pads located at the front of the facility. Four exhaust fans positioned at the opposite end of the house facilitated a consistent unidirectional airflow, enabling the spread of airborne pathogens from the seeder pens across the housing environment. The experimental groups were housed in pens with wood-shaving litter flooring, positioned downstream of the airflow. To minimize the possibility of direct contact and cross-contamination, buffer zones—unoccupied pens—were established between the seeder and treatment pens. The spatial arrangement of pens, airflow direction, and environmental controls were critical to ensuring reliable and consistent exposure across groups, effectively simulating commercial conditions for airborne BCO transmission.

### 2.4. Experimental Design

The experiment included five distinct treatment groups, with each group distributed across six replicate pens. While most groups were reared on wood-shaving litter floors, two pens in Treatment 1 (T1) were maintained on wire flooring to function as seeder birds for initiating the BCO challenge.

Treatment 2 (T2) functioned as the negative control and included six pens with 50 birds per pen, receiving no probiotic supplementation. Treatments 3 (T3), 4 (T4), and 5 (T5) involved the administration of probiotics, either individually or in combination. *GALLIPRO^®^ Hatch* (Novonesis, Hørsholm, Denmark), containing *Enterococcus faecium*, was applied as a spray to day-old chicks in accordance with manufacturer instructions. *GALLIPRO^®^ Fit*, a triple-strain *Bacillus*-based probiotic, was incorporated into the feed at a rate of 500 g per metric ton and provided continuously from day 1 to day 56. The specific composition and assignment of treatments are summarized in Table 1, based on product specifications provided by Novonesis.

All birds received a nutritionally balanced diet formulated to meet commercial production standards in accordance with Cobb 500 recommendations. The diet composition is detailed in Table A1. Feed was manufactured at the University of Arkansas Poultry Research Feed Mill and provided in three phases: starter crumbles from day 0 to day 18, grower pellets from day 18 to day 42, and finisher pellets from day 42 to day 56. Birds had continuous access to feed and clean drinking water throughout the trial.

### 2.5. Lameness Evaluation

Broilers were monitored daily for clinical signs of lameness from day 22 to day 56 of the trial. Birds were gently encouraged to stand and walk, and individuals that showed reluctance to walk, adopted a “goblet gait,” or exhibited signs consistent with spinal deformities such as spondylitis or spondylolisthesis (e.g., sitting on the tail with legs extended, commonly referred to as “kinky back”) were identified as lame. Affected birds were humanely euthanized and subjected to necropsy for further evaluation. Additionally, all mortalities and lameness cases were documented by date and pen, followed by gross pathological examination. Tibial and femoral heads were examined for BCO lesions and scored using the classification criteria described by Wideman [4]. Birds were considered positive for BCO if at least one lesion characteristic of chondronecrosis or osteomyelitis was identified. The cumulative number of lame birds per treatment was determined based on these assessments. Representative images illustrating lesion progression and scoring criteria are presented in Figure 1 and Figure 2.

### 2.6. Body Weight Evaluation

On day 56, the final day of the study, six apparently healthy birds were randomly chosen from each treatment group to measure final body weights and assess for possible subclinical signs of BCO. Tibial and femoral heads were examined and categorized based on the presence and severity of bone lesions.

### 2.7. Statistical Analyses

The cumulative incidence of lameness was calculated for each pen and treatment group using the following formula:*% cumulative lameness per treatment = Number of lame bids per treatment* × 100
*Total number of birds per treatment on D14*

Preliminary comparisons among treatment groups were conducted using the T-test function in Microsoft Excel (Office 365, Microsoft Corp., Redmond, WA, USA). To evaluate the impact of treatments on lameness incidence from day 14 to day 56, logistic regression was applied using a binomial distribution within the generalized linear model (GLM) framework in R (version x64 2.4.2; R Foundation for Statistical Computing, Vienna, Austria). A significance threshold of *p* < 0.05 was used. Data are reported as mean values ± standard error of the mean (SEM).

## 3. Results

Cumulative lameness incidence showed clear differences among treatment groups over the course of the trial (Figure 3). Birds in the positive control (PC) group, reared on wire flooring, exhibited the earliest onset of lameness (beginning around day 33) and the highest cumulative incidence, reaching approximately 77% by day 56. Lameness also developed progressively in the negative control (NC) group, which was housed on litter flooring but exposed to aerosol transmission, culminating in a final incidence of nearly 50%.

In comparison, birds receiving probiotic treatments, T3 (GALLIPRO^®^ Hatch), T4 (GALLIPRO^®^ Fit), and T5 (a combination of Hatch and Fit), exhibited a delayed onset of lameness and significantly lower cumulative lameness levels. By day 56, the final lameness incidence was approximately 31.7% in T3, 31% in T4, and 25.7% in T5, (*p* < 0.05). The T5 group consistently maintained the lowest lameness trajectory throughout the observation period, suggesting a potential synergistic benefit from combining both probiotic treatments. All probiotic interventions significantly reduced lameness compared to the NC group.

Table 2 summarizes the percentage of lame birds with and without mortality across all treatment groups.

Pairwise comparisons of cumulative lameness incidence on day 56 revealed significant differences among treatment groups (Table 3). Birds in the positive control (PC) group exhibited significantly higher odds of developing lameness compared to all probiotic-treated groups: T3 (*p* = *0.0037*), T4 (*p* = *0.0005*), and T5 (*p* = *0.0086*). These reductions suggest that probiotic supplementation, either alone or in combination, markedly decreases lameness risk under BCO challenge conditions. The NC group also had significantly higher cumulative lameness incidence than all treatment groups, with highly significant comparisons to T3 (*p* < *0.0004*), T4 (*p* = *0.0014*), and T5 (*p* < *0.0001*). This further indicates that probiotic supplementation, whether applied individually or in combination, substantially reduced the odds of lameness. When comparing treated groups directly, there was no significant difference between T3 and T4 (*p* = *0.4645*), indicating similar effectiveness. However, T5 exhibited a significantly lower incidence compared to T3 (*p* = *0.0133*) and a marginal difference compared to T4 (*p* = *0.0509*), suggesting a potential additive or synergistic effect when both probiotic products were administered together.

Table 4 presents the weekly cumulative incidence of lameness from day 35 to day 56. Birds in the positive control (PC) group, which were reared on wire flooring, showed the highest lameness levels across all time points, with values increasing from 9.0% on day 35 to 77.0% by day 56. The negative control (NC) group also demonstrated notable lameness progression, rising from 3.0% to 49.0% over the same period. In comparison, birds receiving probiotic treatments (T3, T4, and T5) experienced substantially lower rates of lameness. Specifically, cumulative lameness in T3 increased from 1.0% on day 35 to 31.7% on day 56; T4 ranged from 1.3% to 31.0%; and T5, the group receiving the combined probiotic formulation, maintained the lowest incidence, ranging from 1.7% to 25.7%. The rate of increase in lameness among treated groups was consistently slower than that of both control groups, particularly after day 42. These trends suggest that probiotic supplementation, especially the combined treatment in T5, effectively delayed the onset and reduced the severity of lameness in broilers subjected to a BCO challenge.

Figure 4 shows that administering GALLIPRO^®^ probiotics to broilers significantly reduced cumulative lameness compared to the negative control (NC) group on day 56. Lameness incidence was reduced by 35.4% in T3 (GALLIPRO^®^ Hatch; 31.7% vs. 49.0%), 36.7% in T4 (GALLIPRO^®^ Fit; 31.0% vs. 49.0%), and 47.6% in T5 (GALLIPRO^®^ Hatch/Fit; 25.7% vs. 49.0%), with T5 showing the greatest improvement, (*p* < *0.05*).

Figure 5a,b shows the distribution of femoral and tibial head lesion types among lame birds across all treatment groups. In the femoral head lesion assessment (Figure 5a), normal bone structure (N) in the right femur (RF) was observed in 27.5% of PC birds and 22.5% of NC birds, compared to higher rates in treated groups T3 (17.5%), T4 (20.0%), and T5 (22.5%). A similar pattern was noted in the left femur (LF), where normal femoral heads were reported in 30.0% of PC birds and 25.0% of NC birds, versus 20.0% (T3), 22.5% (T4), and 25.0% (T5). Proximal femoral head separation (FHS) was the most frequent lesion type in both femurs, with right femoral FHS peaking at 22.5% in PC and 20.0% in NC groups. FHS incidence was notably lower in the T3 (10.0%), T4 (15.0%), and T5 (7.5%) groups. Transitional degeneration (FHT) remained under 10% across all treatments, while femoral head necrosis (FHN), a critical indicator of advanced BCO, reached 12.5% in PC birds but dropped to 2.5% in T5.

In tibial head lesion evaluations (Figure 5b), severe necrosis (THNS) was predominant in PC birds, occurring in 65.0% of both the right (RT) and left (LT) tibiae. NC birds showed reduced but still high THNS levels at 37.5% (RT) and 40.0% (LT). In contrast, THNS was markedly lower in probiotic-treated birds: 25.0% (T3 RT), 22.5% (T4 RT), and 17.5% (T5 RT); and 27.5% (T3 LT), 20.0% (T4 LT), and 17.5% (T5 LT). Proximal tibial head necrosis (THN) followed a similar trend, with PC birds showing 10.0% (RT) and 7.5% (LT), while treated groups ranged from 2.5 to 7.5%. Caseous necrosis (THNC) and tibial dyschondroplasia (TD) were rare across all groups, with most values below 5.0%.

In addition, mortality was monitored throughout the trial, with a focus on cumulative deaths from day 29 to day 56. As shown in Figure 6, the positive control (PC) group exhibited the highest cumulative mortality rate at each time point, increasing from 2.0% at day 29 to 11.0% by day 56. The negative control (NC) group followed with a final mortality rate of approximately 6.5%. In contrast, birds receiving probiotic treatments—T3 (*GALLIPRO^®^ Hatch*), T4 (*GALLIPRO^®^ Fit*), and T5 (*Hatch*/*Fit*)—consistently showed lower mortality throughout the study. By day 56, cumulative mortality in T3, T4, and T5 remained below 5%, with T5 recording the lowest overall value.

Table 5 presents the distribution of broiler outcomes, healthy, lame, and dead, expressed as percentages for each treatment group on day 56.

Table 6 presents the mean body weights (±SEM) of clinically healthy birds from each treatment group on day 56. The highest mean weight was observed in T5 (*GALLIPRO^®^ Hatch/Fit*) at 4.76 ± 0.02 kg, followed by T4 (*GALLIPRO^®^ Fit*) at 4.74 ± 0.04 kg, T3 (*GALLIPRO^®^ Hatch*) at 4.72 ± 0.12 kg, and the negative control (NC) at 4.70 ± 0.14 kg. Although treated groups exhibited numerically higher weights, statistical analysis showed no significant differences in mean body weight among the treatments (*p* > 0.05), indicating that probiotic supplementation had no adverse effect on growth performance in healthy birds.

All selected apparently healthy chickens were necropsied to assess clinical lesions (Figure 7). Despite showing no obvious signs of lameness, most of the birds exhibited subclinical signs of lameness across all treatment groups. This suggests that underlying skeletal or musculoskeletal abnormalities may exist, even if the external clinical appearance seems normal.

## 4. Discussion

This study reinforces the potential of probiotics as a practical solution for promoting skeletal integrity and improving overall welfare in broilers challenged with BCO. The BCO model effectively replicated field-relevant conditions, as shown by the sharp and early increase in lameness incidence within the positive control (PC) group, which began to rise around day 33 and reached the highest cumulative rate by day 56. These findings align with prior studies showing that wire flooring exacerbates skeletal loading and tissue stress, which increases the risk of microfractures and facilitates bacterial translocation across a compromised intestinal barrier [23,24,25]. Similarly, the gradual increase in lameness within the negative control (NC) group, which was exposed to aerosolized pathogens on litter flooring, confirms the model’s effectiveness in simulating airborne transmission and mimicking natural outbreak scenarios [14,25].

The treatment groups receiving probiotics—T3 (GALLIPRO^®^ Hatch), T4 (GALLIPRO^®^ Fit), and T5 (combined Hatch/Fit)—all showed marked reductions in cumulative lameness when compared to both control groups (*p* > 0.05). Notably, the T5 group consistently maintained the lowest trajectory of lameness throughout the trial period, indicating potential additive or synergistic benefits when both probiotic formulations were used together. The separation in lameness curves between treatment and control groups became especially evident after day 42, suggesting that the interventions were particularly effective at reducing the progression of BCO during the later growth stages when birds are most vulnerable to lameness. Unlike the PC and NC groups, which experienced steady and significant increases in lameness, all probiotic-treated groups demonstrated delayed onset and a slower rate of cumulative lameness. This suggests that probiotic administration may contribute to improved gut health, enhanced immune responses, and reduced systemic inflammation, which are critical factors in mitigating the onset and severity of BCO.

The observed reductions in lameness among the probiotic-treated groups can be attributed to the multifaceted mechanisms through which these beneficial microbes act. Rather than exerting their effects through a single pathway, strains like *Bacillus subtilis*, *Bacillus amyloliquefaciens*, and *Enterococcus faecium* support broiler health through multiple physiological functions. They help maintain gut homeostasis by stabilizing the microbiota, reinforcing mucosal defenses, promoting the expression of tight junction proteins (e.g., ZO-1, CLDN-2, and OCLN), and modulating immune function [26,27]. Together, these actions enhance the integrity of the intestinal barrier, limiting the translocation of pathogens into systemic circulation. In birds exposed to BCO-inducing conditions, these probiotic-driven improvements likely play a key role in disrupting early disease processes [11,19]. The outcomes of this trial align with previous research demonstrating that *Bacillus*-based supplementation can reduce lameness and help maintain gut architecture under production-related stress [26,27,28].

Our previous studies have also shown that administering *Enterococcus faecium* at hatch contributes to lower BCO-related lameness at market age [14]. Additionally, lactic acid bacteria (LAB) have demonstrated benefits in enhancing intestinal morphology, promoting tighter epithelial junctions, and improving gut permeability under stress conditions [25,28]. Collectively, this evidence emphasizes the value of probiotics not only in supporting gastrointestinal health but also in influencing broader systemic functions such as bone development and immune balance, ultimately contributing to improved animal welfare [11,12].

Analysis of femoral and tibial lesions across treatment groups provided further insight into the complexity of skeletal health in broilers under BCO challenges. Femoral head transitional (FHT) and separation lesions (FHS) were the most prevalent, particularly in the positive control group, while femoral head necrosis (FHN), indicative of advanced pathology, was less frequently observed [3,14,29]. The lower occurrence of FHN may reflect early detection and the ability of some interventions to halt progression before severe damage occurs, consistent with previous findings [23,27,30]. A similar pattern was observed in the tibial head, where necrosis (THN) and its severe form (THNS) were concentrated in the positive control group, aligning with the known vulnerability of tibial growth plates under mechanical and microbial stress [31]. Although birds receiving probiotic treatments showed some reductions in lesion incidence, the lack of a clear trend in lesion severity or distribution across groups supports earlier research suggesting that probiotics may alleviate clinical signs of lameness without fully reversing underlying bone pathology [27]. Lesion data were analyzed descriptively, and no statistical analysis was performed due to the categorical nature of the lesion scores and the limited sample size for some lesion types. Therefore, the observed differences reflect trends rather than statistically confirmed effects.

From a welfare standpoint, both early and advanced skeletal lesions impair mobility. While early lesions such as FHS and FHT may cause discomfort and minor gait changes, more severe conditions like FHN and THNS are linked to significant locomotor impairment, affecting access to feed and water and reducing overall bird welfare [29,31]. These findings point to the importance of early, multifactorial strategies, such as probiotic supplementation, in managing both the visible and underlying components of BCO. Lastly, the combined use of multiple probiotics in this study demonstrated promising outcomes in reducing stress and supporting general health. This integrative approach reflects a growing trend toward sustainable, antibiotic-free strategies in poultry production systems. While this manuscript focuses on lameness incidence and performance outcomes, additional data from this study will be presented in a forthcoming companion publication. These include microbiological cultures, 16S rRNA sequencing of gut and bone microbiota, host immune gene expression, and tight junction gene expression (e.g., OCLN, CLDN-2, ZO-1), which together provide deeper insight into the microbial, immunological, and epithelial mechanisms underlying BCO development and the effects of probiotic interventions.

## 5. Conclusions

This study demonstrated that administering *Enterococcus faecium* at hatch and a triple-strain, Bacillus-based feed additive significantly reduced BCO-related lameness in broilers under an aerosol transmission challenge. Both treatments, when used individually, reduced lameness by ~35–37%, while their combination achieved the greatest reduction (~48%), suggesting a synergistic effect. These findings highlight the potential of targeted probiotic strategies as sustainable, antibiotic-free tools to improve skeletal health and welfare in broiler chickens. Further investigation into the underlying mechanisms, such as modulation of gut microbiota, immune responses, and bacterial translocation, will provide greater insight into their protective effects. Validating these results under commercial production conditions is essential to confirm their practical application and scalability. Overall, this study contributes to ongoing efforts to enhance poultry health, welfare, and sustainability in modern broiler production systems.

## Figures and Tables

**Figure 1 animals-15-01765-f001:**
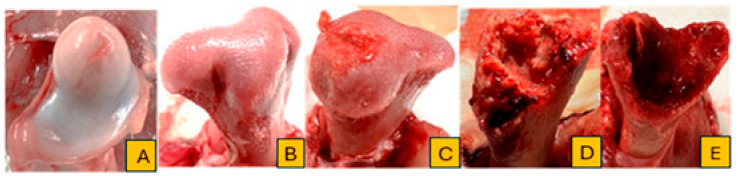
Classification of femoral head lesions used to evaluate BCO progression: (**A**) normal; (**B**) femoral head separation (FHS); (**C**) femoral head transitional degeneration (FHT); (**D**,**E**) femoral head necrosis (FHN) (Antheny et al. [14]).

**Figure 2 animals-15-01765-f002:**
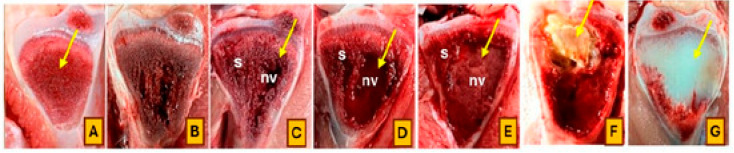
Categories of tibial head lesions associated with the progression of BCO: (**A**) normal; (**B**) tibial head necrosis; (**C**–**E**) severe tibial head necrosis; (**F**) caseous tibial head necrosis; (**G**) tibial dyschondroplasia; (s) sequestrae, (nv) necrotic voids, and microfractures below the growth plate (arrows). (Antheny et al. [14]).

**Figure 3 animals-15-01765-f003:**
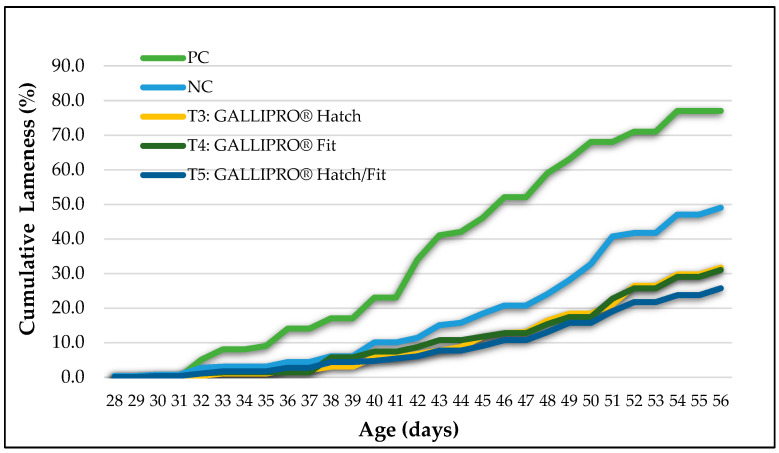
Cumulative lameness percentages by treatment group from day 28 to day 56 of the study. PC = positive control, NC = negative control, T3 = GALLIPRO^®^ Hatch (1.25 mL/chick spray vaccination on d0), T4 = GALLIPRO^®^ Fit (at 500 g/t feed), and T5 = GALLIPRO^®^ Hatch/Fit.

**Figure 4 animals-15-01765-f004:**
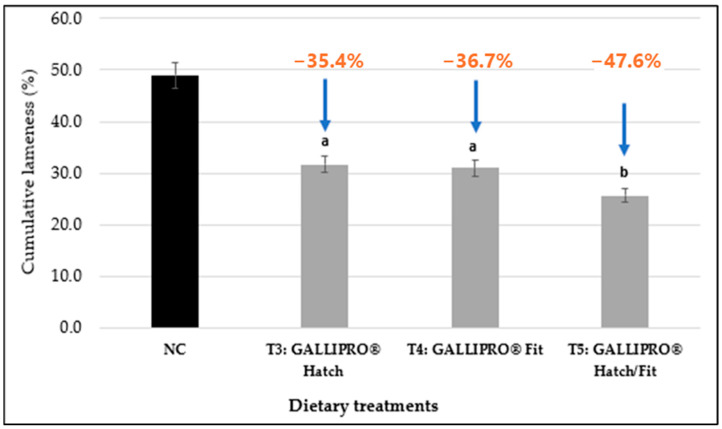
Evaluation of the impact of GALLIPRO^®^ probiotic treatments (T3, T4, and T5) on lameness reduction compared to the negative control (NC) group. Different superscript letters (a, b) within the same category denote statistically significant differences (*p* < 0.05). Error bars indicate the standard error of the mean (SEM). PC = positive control, NC = negative control, T3 = GALLIPRO^®^ Hatch (1.25 mL/chick spray vaccination on d0), T4 = GALLIPRO^®^ Fit (at 500 g/t feed), and T5 = GALLIPRO^®^ Hatch/Fit.

**Figure 5 animals-15-01765-f005:**
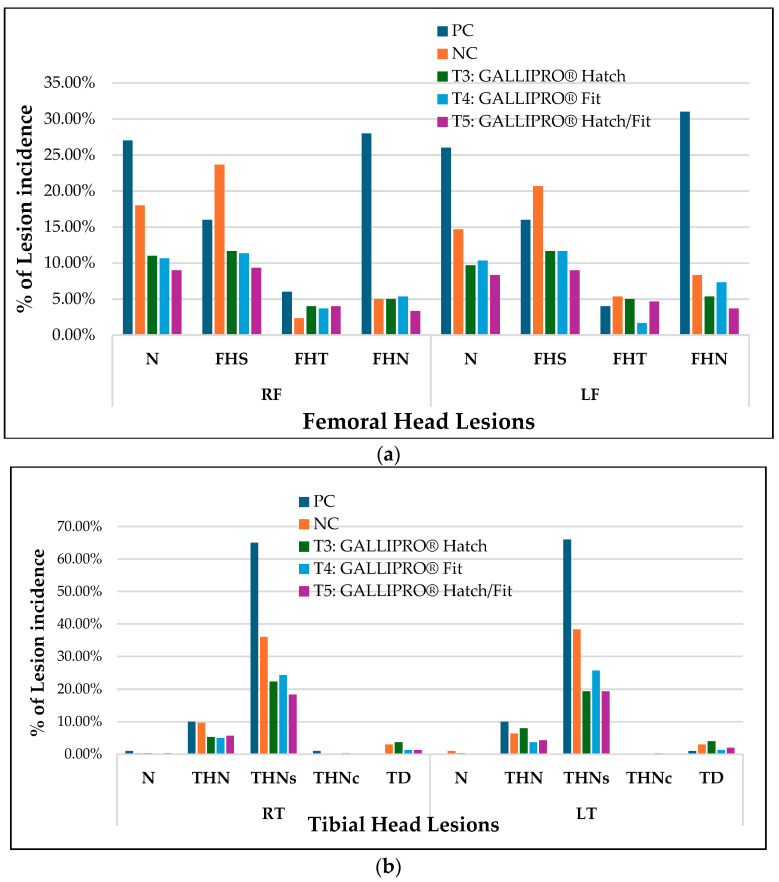
(**a**): Femoral lesion ratings for birds diagnosed as lame. The percentage of lame birds with the indicated BCO lesion(s) is plotted for each bone by treatment. RF—right femur, LF—left femur, N = femur head and proximal tibia appear entirely normal, FHS = proximal femoral head separation (epiphyseolysis), FHT = proximal femoral head transitional degeneration, FHN = proximal femoral head necrosis. (Bacterial chondronecrosis with osteomyelitis, BCO). PC = positive control, NC = negative control, T3 = GALLIPRO^®^ Hatch (1.25 mL/chick spray vaccination on d0), T4 = GALLIPRO^®^ Fit (at 500 g/t feed), and T5 = GALLIPRO^®^ Hatch/Fit. (**b**): Tibial lesion scores among birds identified as lame. The graph displays the percentage of lame birds exhibiting specific BCO-related lesions in the right (RT) and left (LT) tibiae across treatment groups. Lesion types include THN—proximal tibial head necrosis; THNC—caseous proximal tibial head necrosis; THNS—severe proximal tibial head necrosis; and TD—tibial dyschondroplasia. Treatment groups are defined as follows: PC—positive control; NC—negative control; T3—GALLIPRO^®^ Hatch (spray vaccination at 1.25 mL per chick on day 0); T4—GALLIPRO^®^ Fit (500 g/ton feed inclusion); and T5—combination of GALLIPRO^®^ Hatch and Fit.

**Figure 6 animals-15-01765-f006:**
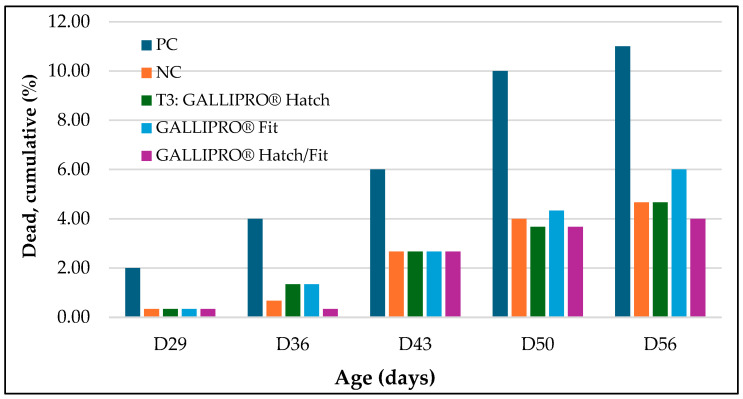
Weekly cumulative mortality (%) recorded from day 29 to day 56 across all treatment groups. Mortality encompasses all recorded causes of death during the study period. Treatment groups are defined as follows: PC—positive control; NC—negative control; T3—GALLIPRO^®^ Hatch (administered as a 1.25 mL/chick spray on day 0); T4—GALLIPRO^®^ Fit (included in feed at 500 g/ton); T5—combination of GALLIPRO^®^ Hatch and Fit.

**Figure 7 animals-15-01765-f007:**
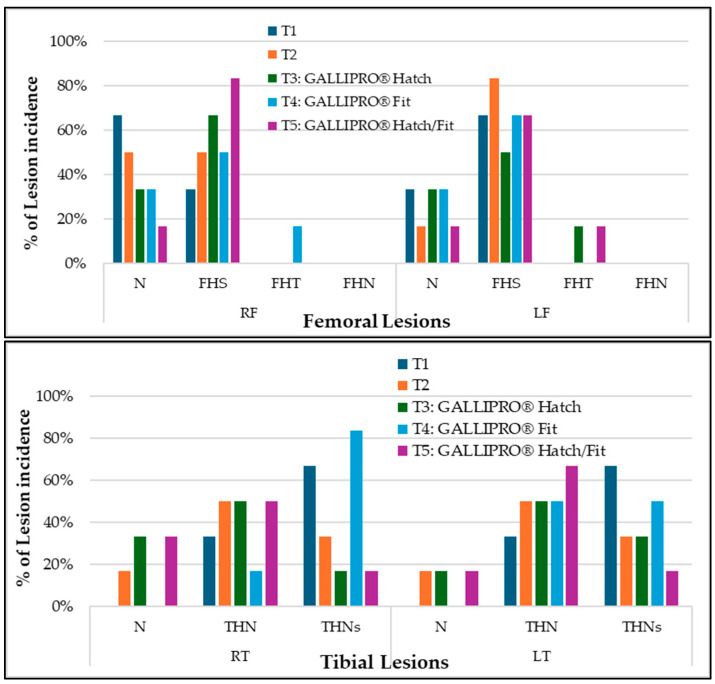
Distribution of femoral (right and left) and tibial (right and left) lesion types in clinically healthy broilers. Lesion categories include normal (N), femoral head separation (FHS), femoral head transitional degeneration (FHT), femoral head necrosis (FHN), proximal tibial head necrosis (THN), and severe tibial head necrosis (THNs), PC = positive control, NC = negative control, T3 = GALLIPRO^®^ Hatch (1.25 mL/chick spray vaccination on d0), T4 = GALLIPRO^®^ Fit (at 500 g/t feed), and T5 = GALLIPRO^®^ Hatch/Fit.

**Table 1 animals-15-01765-t001:** Overview of treatment group assignments used in the study.

Treatment Group	Floor Type	Number of Pens Assigned
T1: Positive Control—Infection source	Wire	2
T2: Negative Control	Litter	6
T3: GALLIPRO^®^ Hatch (1.25 mL/chick spray vaccination on d0)	Litter	6
T4: GALLIPRO^®^ Fit (at 500 g/t feed)	Litter	6
T5: GALLIPRO^®^ Hatch/Fit	Litter	6

**Table 2 animals-15-01765-t002:** The percentage of lame birds with and without mortality across all treatment groups.

	PC	NC	T3	T4	T5
% Lame birds with deaths	77.0	49.0	31.7	31.0	25.7
Total death	4	12	10	11	13
% Lame birds without deaths	66.0	44.3	27.0	25.0	21.7

PC = positive control, NC = negative control, T3 = GALLIPRO^®^ Hatch (1.25 mL/chick spray vaccination on d0), T4 = GALLIPRO^®^ Fit (at 500 g/t feed), and T5 = GALLIPRO^®^ Hatch/Fit.

**Table 3 animals-15-01765-t003:** Pairwise comparison of cumulative lameness incidence at d56 between treatment groups using the *T*-test function in Excel.

*p*-Value	NC	T3	T4	T5
**PC**	0.0036 *	0.0037 *	0.0005 *	0.0086 *
**T2**		<0.0004 *	0.0014 *	<0.0001 *
**T3**			0.4645	0.0133 *
**T4**				0.0509

*Note:* An asterisk (*) denotes a statistically significant difference (*p* < 0.05). Treatment groups are defined as follows: PC = positive control, NC = negative control, T3 = GALLIPRO^®^ Hatch (1.25 mL/chick spray vaccination on d0), T4 = GALLIPRO^®^ Fit (at 500 g/t feed), and T5 = GALLIPRO^®^ Hatch/Fit.

**Table 4 animals-15-01765-t004:** Weekly cumulative lameness progression from day 35 to day 56, expressed as percentages.

Day	PC	NC	T3	T4	T5
35	9.0	3.0	1.0	1.3	1.7
42	34.0	11.3	6.7	8.7	6.0
49	63.0	28.0	18.3	17.3	15.7
56	77.0	49.0	31.7	31.0	25.7

**Note:** Treatment groups are designated as follows: PC—positive control; NC—negative control; T3—GALLIPRO^®^ Hatch (spray-applied at 1.25 mL per chick on day 0); T4—GALLIPRO^®^ Fit (included in feed at 500 g/ton); and T5—combination of GALLIPRO^®^ Hatch and Fit. The table provides descriptive weekly lameness data without statistical analysis at individual time points; thus, *p*-values and SEMs are not applicable. Cumulative statistical comparisons are presented in Table 3.

**Table 5 animals-15-01765-t005:** Percentage of healthy birds, lameness incidence, and cumulative mortality at day 56 across treatment groups.

Condition	PC	NC	T3	T4	T5
KB	2.0	0.0	0.0	0.0	0.7
DUR	6.0	2.0	1.0	2.7	1.3
SDS	5.0	2.3	3.3	3.3	2.7
LAME	77.0 ^a^	49.0 ^b^	31.7 ^c^	31.0 ^c^	25.7 ^c^
HEALTHY	10.0 ^g^	46.7 ^f^	64.0 ^e^	63.0 ^e^	69.6 ^d^

Note: KB = Kinky back (a form of lameness); DUR = Death due to unknown causes; SDS = Sudden Death Syndrome. Treatment groups are defined as follows: PC—positive control; NC—negative control; T3—GALLIPRO^®^ Hatch (spray applied at 1.25 mL per chick on day 0); T4—GALLIPRO^®^ Fit (included in feed at 500 g/ton); T5—combination of GALLIPRO^®^ Hatch and Fit. Data are expressed as categorical percentages and were analyzed using one-way ANOVA followed by Tukey’s post hoc test. Different superscript letters (a–g) within rows indicate statistically significant differences (*p* < 0.05). SEM is not applicable for categorical percentage values.

**Table 6 animals-15-01765-t006:** Mean body weights (kg) of clinically healthy birds across all treatment groups on day 56.

Treatment	Number of Birds	Average Weight (±SEM, in kg)
NC	6	4.70 ± 0.14
T3	6	4.72 ± 0.12
T4	6	4.74 ± 0.04
T5	6	4.76 ± 0.02

**Note:** Mean body weights did not differ significantly among the four treatment groups (*p* > 0.05). Abbreviations: PC—positive control; NC—negative control; T3—GALLIPRO^®^ Hatch (spray vaccination at 1.25 mL per chick on day 0); T4—GALLIPRO^®^ Fit (included in feed at 500 g/ton); T5—combination of GALLIPRO^®^ Hatch and Fit.

## Data Availability

The data generated and/or analyzed during this study can be obtained from the corresponding author (A.A.K.A.) upon reasonable request.

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
