# Peer review of "Assessing the Impact of Spraying an E. faecium Probiotic at Hatch and Supplementing Feed with a Triple-Strain Bacillus-Based Additive on BCO Lameness Incidence in Broiler Chickens"

_animals, 2025, doi:10.3390/ani15121765_

Round 1
Reviewer 1 Report
Comments and Suggestions for Authors
The manuscript is well written and authors have demonstrated the effectiveness of probiotics in reducing BCO in broilers. Manuscript should be improved b answering the following comments and by doing little bit of extra laboratory work.
1. Line 42. 'Greatest improvement' is not a technical term. Use the statistically highest if it was. Also, the combined effect was better than single treatment, but it is too synergistic, not additive.
2. Line 48-63. In text citation is (1-3) which is not justified. Break the citation where appropriate and after each sentence or each fact, cite it. Just citing at the end of paragraphs is not good. Do the same for whole manuscript, some of the citations are incorrect as well.
3. In the introduction section, add possible mechanisms and the molecular basis of benefits of probiotics in BCO. Also add relevant studies conducted previously.
4. There is very only a small difference in commutative lamness in PC and NC groups. How do the authors justify this. Also explain in more details, how thee BCO was induced, was any special pathogen used.
5. Since the authors argued about infectious causes of BCO, why microbiology of the blood or bone is not included in the study. Augment the study results with direct microbiology findings, or metagenomic analysis of gut/bones, blood in different groups or quantitative enumeration of specific groups of microbes in gut/blood/bone.
6. Authors have used only the symptomatic results and have not tried to elucidate any specific mechanisms. Also, augment the data by adding the expression of relevant genes.
7. The discussion part is all over the place, make it coherent, also the conclusion is too long.
In discussion part, explain the molecular mechanisms and
Author Response
May 25, 2025
Dear reviewer,
Thank you for having taken the time to review the manuscript and provide us with invaluable feedback. Please find our addressing of your feedback to the original manuscript below:
Comment 1. Line 42. 'Greatest improvement' is not a technical term. Use the statistically highest if it was. Also, the combined effect was better than single treatment, but it is too synergistic, not additive.
Respond 1: Addressed.
Comment2. Line 48-63. In text citation is (1-3) which is not justified. Break the citation where appropriate and after each sentence or each fact, cite it. Just citing at the end of paragraphs is not good. Do the same for whole manuscript, some of the citations are incorrect as well.
Respond 2: Done.
Comment 3. In the introduction section, add possible mechanisms and the molecular basis of benefits of probiotics in BCO. Also add relevant studies conducted previously.
Respond 3: Added between lines 87-97
Comment 4. There is very only a small difference in commutative lamness in PC and NC groups. How do the authors justify this. Also explain in more details, how thee BCO was induced, was any special pathogen used.
Respond 4: We appreciate the reviewer’s observation regarding the relatively small difference in cumulative lameness between the positive control (PC) and negative control (NC) groups. We respectfully clarify that this outcome reinforces the effectiveness of the BCO-induced aerosol transmission model used in our study. As detailed in the manuscript (Section 2.3), BCO was induced through a wire-floor aerosol challenge setup, wherein two pens of birds (PC group) served as the infection source. The directional airflow distributed airborne pathogens to the surrounding pens (including NC), thereby mimicking natural environmental exposure observed in commercial settings. This model is supported by previous publications [Asnayanti et al., 2024; Wideman et al., 2013], and the gradual onset of lameness in the NC group further validates its field relevance.
No special or externally introduced pathogens were used. The model relied on natural opportunistic bacteria present in the environment, particularly those commonly associated with BCO (e.g., Enterococcus, Staphylococcus, and Escherichia coli), as noted in our abstract and introduction. This approach mirrors real-world production systems, where BCO develops under the influence of mechanical stress, intestinal translocation, and environmental challenge rather than direct inoculation with specific pathogens.
Comment 5. Since the authors argued about infectious causes of BCO, why microbiology of the blood or bone is not included in the study. Augment the study results with direct microbiology findings, or metagenomic analysis of gut/bones, blood in different groups or quantitative enumeration of specific groups of microbes in gut/blood/bone.
Respond 5: Microbiological, molecular, and immunological analyses were performed as part of this research; however, due to the breadth of the dataset, the study was divided into two publications. The current manuscript focuses on cumulative lameness incidence and performance outcomes. A second manuscript, currently in preparation, will report findings from culture-based pathogen isolation, 16S rRNA sequencing of gut and bone microbiota, and host immune gene expression profiling across treatment groups. This approach allowed for a more detailed and focused presentation of each dataset while maintaining scientific clarity and coherence. A note has been added to the Discussion to inform readers of the forthcoming companion paper and added bettween lines: 488-495.
Comment 6. Authors have used only the symptomatic results and have not tried to elucidate any specific mechanisms. Also, augment the data by adding the expression of relevant genes.
Respond 6: Please refer to our response to Comment 5, which addresses the microbiological and mechanistic analyses associated with this study.
Comment7. The discussion part is all over the place, make it coherent, also the conclusion is too long.
Respond 7: Addressed.
Reviewer 2 Report
Comments and Suggestions for Authors
The manuscript has been evaluated in detail.
As a result of the examination, it was understood that the effects of bacillus and enterococcus probiotic bacteria on reducing the incidence of bacterial chondronecrosis with osteomyelitis (BCO) were investigated and that probiotics prevented the formation of BCO and improved the skeletal system of chickens.
The results obtained in terms of the topic of the article and the improvement of animal welfare are very valuable. The corrections that need to be made in the article are given below.
In lines 18 and 58, there is a positive verb used in a negative sentence. It would be better to use the verb cause.
The sentence in lines 87-88 is not suitable for the introduction. While mentioning the benefits of probiotics in the introduction, mentioning enteric infection of C perfingens distorts the meaning of the paragraph. It should be deleted.
Reference number 14 in line 87 should be a study on the benefits of E faecium, but reference number 14 includes the benefits of Bacillus. It is not a correct reference. The correct reference should be added or deleted.
In line 100, reference 21 does not contain information on BCO or skeletal development, I could only find information on necrotic enteridis in the intestines. Reference 21 is not correct for this article.
Line 195 should be Wideman [3]
Line 2008 reference Antheny et al., (2025) should be either [20] or Antheny et al. [20]
Line 262 table title should be corrected, the title should be completed by deleting the space in between
P values ​​are given in table 3, but not in other tables. SEM values ​​are not given in table 4. SEM values ​​are not given in table 5.
All literature in the article should be revised according to MDPI writing rules
Author Response
May 25, 2025
Dear reviewer,
Thank you for having taken the time to review the manuscript and provide us with invaluable feedback. Please find our addressing of your feedback to the original manuscript below:
Comment 1. In lines 18 and 58, there is a positive verb used in a negative sentence. It would be better to use the verb cause.
Respond 1: Addressed by using verb “Cause”.
Comment 2. The sentence in lines 87-88 is not suitable for the introduction. While mentioning the benefits of probiotics in the introduction, mentioning enteric infection of C perfingens distorts the meaning of the paragraph. It should be deleted.
Respond 2: Deleted it.
Comment 3. Reference number 14 in line 87 should be a study on the benefits of E faecium, but reference number 14 includes the benefits of Bacillus. It is not a correct reference. The correct reference should be added or deleted.
Respond 3: Addressed.
Comment 4. In line 100, reference 21 does not contain information on BCO or skeletal development, I could only find information on necrotic enteridis in the intestines.
Respond 4: Addressed.
Comment 5. Reference 21 is not correct for this article.
Respond 5: Addressed.
Comment 6. Line 195 should be Wideman [3].
Respond 6: Addressed.
Comment 7. Line 2008 reference Antheny et al., (2025) should be either [20] or Antheny et al. [20].
Respond 7: Addressed.
Comment 8. Line 262 table title should be corrected, the title should be completed by deleting the space in between
Respond 8: Addressed.
Comment 9. P values ​​are given in table 3, but not in other tables. SEM values ​​are not given in table 4. SEM values ​​are not given in table 5.
Respond 9: Regarding Table 3, P-values were included to highlight statistically significant differences from pairwise comparisons of cumulative lameness incidence using T-tests. However, in Table 4, the weekly progression of lameness is presented as descriptive trend data rather than as a statistical comparison at each time point. Since the primary statistical analysis was conducted on cumulative outcomes on day 56 (as shown in Table 3), inferential statistics were not repeated for each week in Table 4.
Similarly, Table 5 presents categorical outcome data at the end of the study (e.g., percentages of healthy, lame, and dead birds) and was analyzed using ANOVA followed by Tukey’s post hoc test. Significant differences are indicated by superscripts within rows, as noted in the table. Therefore, individual SEM values are not included, as the data were not structured for standard deviation-based reporting. Including SEMs would not be statistically meaningful in this context and could misrepresent the nature of the analysis. We respectfully believe that the tables, as presented, appropriately reflect the type of data and analysis conducted. Nonetheless, we have clarified this rationale in the table legends to guide readers accordingly and added footnotes under each table.
Comment 10. All literature in the article should be revised according to MDPI writing rules
Respond 10: Addressed.
Reviewer 3 Report
Comments and Suggestions for Authors
The following manuscript, titled “Assessing the Impact of Spraying an E. faecium Probiotic at Hatch and Supplementing Feed with a Triple-Strain Bacillus-Based Additive on BCO Lameness Incidence in Broiler Chickens,” investigates the effect of different probiotics on the incidence of BCO lameness in chickens. This is a well-written manuscript; however, there are a few issues that need to be addressed before it can move forward.
- Abstract: The introductory section is too long. Please decrease it to 2–3 sentences and expand the sections on results, discussion, and conclusion to better reflect the main findings and significance of the study
- Introduction: Please add more references to support the statements made, particularly around lines 49, 53, 54, 58, 85, 92, and 96.
- Results: In all figures, please include detailed descriptions of the treatments. For example, instead of 'T3: GALLIPRO® Hatch', use 'T3: GALLIPRO® Hatch (1.25 mL/chick spray vaccination on d0)'. Apply the same approach to T4 and T5. This will allow each figure to stand on its own without requiring readers to refer back to the materials and methods section
- Did you perform statistical analysis for Table 5, Figures 5a and 5b, and Figures 7a and 7b? If so, please indicate the statistical tests used and include relevant significance letters in the table and figures. In case it is not significant, please mention that in the figure footnote.
- Discussion: Line 395: “compared to both control groups,” add P-value in this sentence.
- Line 413-414: add a reference that supports this claim: “limiting translocation of pathogenic bacteria”
- Line 424-426: add a reference that supports this claim.
- In the discussion, anytime a claim is made, it should be followed by a reference. Please fix this issue in the discussion section.
- If statistical analysis was done on parameters discussed between lines 428-lines 441, please add the p-values.
- Remove all “-” in the introduction, discussion, and conclusion sections. Example: Line 404-405: “enhanced immune responses and reduced systemic inflammation—factors that are critical in mitigating the onset and severity of BCO.”
Author Response
May 25, 2025
Dear reviewer,
Thank you for having taken the time to review the manuscript and provide us with invaluable feedback. Please find our addressing of your feedback to the original manuscript below.
Comment 1. Abstract: The introductory section is too long. Please decrease it to 2–3 sentences and expand the sections on results, discussion, and conclusion to better reflect the main findings and significance of the study.
Respond 1: Addressed.
Comment 2. Introduction: Please add more references to support the statements made, particularly around lines 49, 53, 54, 58, 85, 92, and 96.
Respond 2: Addressed.
Comment 3. Results: In all figures, please include detailed descriptions of the treatments. For example, instead of 'T3: GALLIPRO® Hatch', use 'T3: GALLIPRO® Hatch (1.25 mL/chick spray vaccination on d0)'. Apply the same approach to T4 and T5. This will allow each figure to stand on its own without requiring readers to refer back to the materials and methods section.
Respond 3: Addressed.
Comment 4. Did you perform statistical analysis for Table 5, Figures 5a and 5b, and Figures 7a and 7b? If so, please indicate the statistical tests used and include relevant significance letters in the table and figures. In case it is not significant, please mention that in the figure footnote.
Respond 4: Yes, statistical analysis was performed for the data presented in Table 5. Specifically, lameness incidence and the percentage of healthy birds across treatment groups were analyzed using one-way ANOVA followed by Tukey’s post hoc test. The corresponding p-values have now been added to the Results section, and significant differences are indicated in Table 5. However, statistical analysis was not conducted for Figures 5a, 5b, 7a, and 7b, as these figures were intended to provide descriptive visual trends rather than inferential comparisons.
Comment 5. Discussion: Line 395: “compared to both control groups,” add P-value in this sentence.
Respond 5: Addressed.
Comment 6. Line 413-414: add a reference that supports this claim: “limiting translocation of pathogenic bacteria”
Respond 6: Addrssed.
Comment 7. Line 424-426: add a reference that supports this claim.
Respond 7: Addressed.
Comment 8. In the discussion, anytime a claim is made, it should be followed by a reference. Please fix this issue in the discussion section.
Respond 8: Done.
Comment 9. If statistical analysis was done on parameters discussed between lines 428-lines 441, please add the p-values.
Respond 9: Statistical analysis was not performed for the lesion distribution data discussed between lines 428–441. These observations were intended to provide a descriptive summary of pathological findings rather than inferential comparisons. The frequency and severity of specific lesions (e.g., FHS, FHT, FHN, THN, THNS) varied across groups but did not meet the criteria for statistical testing due to categorical scoring limitations and low sample sizes within lesion subtypes. We have clarified this in the revised text and added a statement noting that these data were analyzed descriptively without statistical testing.
Added between lines: 470-473.
Comment 10. Remove all “-” in the introduction, discussion, and conclusion sections. Example: Line 404-405: “enhanced immune responses and reduced systemic inflammation—factors that are critical in mitigating the onset and severity of BCO.”
Respond 10: Addressed.
Round 2
Reviewer 1 Report
Comments and Suggestions for Authors
It is an improved version now.
Reviewer 3 Report
Comments and Suggestions for Authors
Thank you for addressing my comments thoroughly and responding to each point.